# Biodiesel-Derived Glycerol Obtained from Renewable Biomass—A Suitable Substrate for the Growth of *Candida zeylanoides* Yeast Strain ATCC 20367

**DOI:** 10.3390/microorganisms7080265

**Published:** 2019-08-16

**Authors:** Laura Mitrea, Floricuța Ranga, Florinela Fetea, Francisc Vasile Dulf, Alexandru Rusu, Monica Trif, Dan Cristian Vodnar

**Affiliations:** 1Institute of Life Sciences, Faculty of Food Science and Technology, University of Agricultural Sciences and Veterinary Medicine, Cluj-Napoca, Calea Mănăștur 3-5, 400372 Cluj-Napoca, Romania; 2Faculty of Agriculture, University of Agricultural Sciences and Veterinary Medicine Cluj-Napoca, Calea Mănăștur 3-5, 400372 Cluj-Napoca, Romania; 3CENCIRA Agrofood Research and Innovation Centre, 6/66 Ion Meșter Street, 400650 Cluj-Napoca, Romania

**Keywords:** renewable biomass, used kitchen oil, crude glycerol, organic acids, *Candida zeylanoides*, fermentations

## Abstract

Used kitchen oil represents a feasible and renewable biomass to produce green biofuels such as biodiesel. Biodiesel production generates large amounts of by-products such as the crude glycerol fraction, which can be further used biotechnologically as a valuable nutrient for many microorganisms. In this study, we transesterified used kitchen oil with methanol and sodium hydroxide in order to obtain biodiesel and crude glycerol fractions. The crude glycerol fraction consisting of 30% glycerol was integrated into a bioreactor cultivation process as a nutrient source for the growth of *Candida zeylanoides* ATCC 20367. Cell viability and biomass production were similar to those obtained with batch cultivations on pure glycerol or glucose as the main nutrient substrates. However, the biosynthesis of organic acids (e.g., citric and succinic) was significantly different compared to pure glycerol and glucose used as main carbon sources.

## 1. Introduction

Green biofuels have gained the attention of researchers in the last decades, not only because of the imminent exhaustion of the fossil fuels, but also due to the renewable biomass’ potential to be converted into bio-combustibles with high efficiency [1,2]. Biofuels, and particularly biodiesel, can be successfully produced from biomass through catalytic reactions [3,4,5]. Biodiesel, which is an important exponent of eco-friendly biofuels, is produced in large quantities around the globe and especially in Europe, which is the leader in the context of biodiesel production market [6,7]. One of the major advantages of biodiesel is given by its non-toxicity and its minimal greenhouse gas emissions [8].

Biomass is presented as a renewable resource for bioenergy and biochemical production [9,10]. It mostly consists of wood wastes, agricultural crops and their waste derivatives, municipal solid wastes, animal wastes, and residues deriving from food and aquatic plant processing industries; all these can be sources of biogenic and renewable biomaterials and biofuels [9,11,12,13,14,15,16]. From the economical point of view, recycled oils and greases resulting from the food sector represent a feasible alternative source of renewable biomass for the biodiesel industry [9,17]. Massive quantities of used kitchen oils and greases are generated worldwide, and in technologically-advanced countries in particular, their disposal is causing important environmental issues [18]. Reusing the used kitchen oils via biotechnological processes with the purpose of producing biofuels, biomaterials, and biochemicals is a step forward in the reduction of environmental pollution caused by their random discharge [19]. Moreover, using recycled kitchen oils in the biofuels industry as feedstock material instead of edible oils (e.g., canola, soybean, sunflower, palm oils) might avoid the competition between the exploitation of lands for food versus energy [19,20,21,22].

The triglycerides from vegetal or animal sources can be efficiently converted into biodiesel through a variety of procedures, such as direct mixtures with solvents, micro-emulsions, pyrolysis, and transesterification [23,24,25,26]. The biodiesel phase consisting of fatty acids methyl/ethyl esters is mostly achieved through transesterification using various catalysts [27]. The transesterification reaction and the quality of the final products depend significantly on the catalyst type (alkaline (sodium hydroxide, sodium methoxide, potassium hydroxide) vs. acidic (sulfuric acid, sulphonic acid, hydrochloric acid) vs. enzymatic (lipase)) and its concentration [20,24,28,29,30]. During transesterification, the ester bonds between fatty acids and glycerol break down, and the free fatty acids bind to other alcohol molecules present in the reaction medium. At the end of the transesterification process, the reaction mixture contains an amalgam of fatty acids, methyl/alkyl esters, and glycerol [31]. Regardless of the procedure used in the manufacturing of biodiesel, extensive amounts of by-products (e.g., crude glycerol) are generated, and the surplus of this fraction presents environmental challenges associated with its disposal [19,32,33].

In the biotechnological context, crude glycerol represents a valuable matrix that can be exploited for the bio-production of chemicals with health [5,34] or industrial potential [26,35,36,37,38]. Bacteria, fungi, yeasts, and algae can metabolize biodiesel-derived crude glycerol and convert it into organic acids, propane-diols, carotenes, poly-unsaturated fatty acids, proteins, lipids, etc. [5,26,34,38,39]. Fungal and yeast strains such as *Aspergillus*, *Rhizopus*, *Yarrowia* or *Candida* are mostly cultivated for their ability to transform the crude glycerol fraction into bio-chemicals with food applicability: organic acids (citric, succinic, malic acids), low caloric polyols (arabitol, erythritol, mannitol), or single-cell oils [40]. Multiple studies have been conducted on crude glycerol using yeasts such as *Candida* or *Yarrowia* to produce feedstock chemicals such as citric and succinic acids [41] under steady-state conditions and under different grades of aeration [42,43]. Some of the *Yarrowia* and the *Candida* species (*Yarrowia lipolytica*, *Candida tropicalis*, *Candida guilliermondii*, *Candida parapsilopsis*, *Candida oleophila*, *Candida zeylanoides*) are well-known as adaptable microorganisms to fats, oils, fatty acids or hydrocarbons containing medium [40,43] because of their ability to metabolize hydrophobic substrates [44,45].

In the context of pollution reduction, crude glycerol derived from biodiesel production using recycled kitchen oils represents a suitable medium for cultivating microorganisms such as *Candida* spp. The main purpose of this work was to evaluate the adaptation mechanisms of *C. zeylanoides* ATCC 20367 cells to a cultivation medium that contains only crude glycerol as an energy source and their potential to bio-synthesize citric and succinic acids. Crude glycerol obtained from recycled kitchen oil by alkali transesterification was used as a single energy substrate, and analytical grade glycerol and glucose were used as comparators for cell viability and metabolites production.

## 2. Materials and Methods

### 2.1. Materials 

Except for crude glycerol fraction, all the culture media components were of analytical grade and purchased from VWR International (Radnor, Pennsylvania, PA, USA).

The crude glycerol fraction was obtained from recycled vegetable cooking oil through an alkali transesterification process. The fried sunflower oil collected from the household was mixed with methanol and NaOH. The inferior phase consisting of crude glycerol was collected after 48 h [46] and used as a nutrient substrate for batch fermentations. Before the addition to fermentation medium, the pH of the crude glycerol fraction was adjusted to 7 by adding a few drops of 2M HCl, and it was measured with a laboratory pH meter, model InoLab. The crude glycerol fraction was sterilized separately at 121 °C for 20 min.

### 2.2. Microorganism and Culture Media

*C. zeylanoides* ATCC 20367 purchased from American Type Culture Collection (Manassas, Virginia, VA, USA) was used in the present work for all the experiments. The yeast strain was maintained on yeast malt extract agar plates (at 1 L distilled water: yeast extract 3 g/L, malt extract 3 g/L, dextrose 10 g/L, peptone 5 g/L, agar 20 g/L) at 4 °C and renewed periodically every 2–3 months.

Culture media components and cultivation conditions were adapted after Takayama et al. [47]. The inoculums representing 10% of the culture were prepared by transferring 10^7^ cfu/mL into 500 mL shake flasks with buffers containing 200 mL of culture media having the components mentioned in Table 1. The shake flasks were incubated for 2 days at 30 °C, pH 6 ± 0.2, and 200 rpm. The fermentation process conducted at the bioreactor level contained the same components except for CaCO_3_.

### 2.3. Bioreactor Batch Fermentation

All experiments were performed in a 5 L bioreactor (Eppendorf, BioFlo 320, one unit, Hamburg, Germany) containing 2 L of working media. The inoculums were added in sterile conditions. The bioreactor was equipped with pH and temperature sensors and a rotation speed control. Temperature, pH, and rotations were maintained constant at 30 °C, 6.00 ± 0.2, and 400 rpm, respectively. The pH was adjusted automatically by adding 45% NaOH. The fermentation process ran for 163 h in aerobic conditions. Filtered air (through 0.20 µm filters, Macherey-Nagel) was continuously added into fermentation broth by a peristaltic pump (Watson Marlow 520 S, Cornwall, England) settled at 10 rpm and 158 mL/min. From time to time, sterilized silicone oil was added as an antifoaming agent. Samples were collected to perform specific tests at regular time intervals.

### 2.4. Assays

#### 2.4.1. Determination of the Fatty Acids from Processed and Unprocessed Vegetable Oil

The fatty acids content from vegetable oil before and after processing within the kitchen was analyzed by GC. The fatty acids profile of the total lipids was determined by acid-catalyzed transesterification by using 1% sulphuric acid in methanol [48,49]. The methylated fatty acids were determined with a gas chromatograph coupled to a mass spectrometer (model PerkinElmer Clarus 600 T GC-MS; PerkinElmer, Inc., Shelton, CT, USA) [50]. A 0.5 μL sample was injected into a 60 m × 0.25 mm i.d., 0.25 μm film thickness SUPELCOWAX 10 capillary column (Supelco Inc., Darmstadt, Germany). The operation conditions were as follows: injector temperature 210 °C; helium carrier gas flow rate 0.8 mL/min; split ratio 1:24; oven temperature 140 °C (hold 2 min) to 220 °C at 7 °C/min (hold 23 min); electron impact ionization voltage 70 eV; trap current 100 μA; ion source temperature 150 °C; mass range 22−395 m/z (0.14 scans/s with an intermediate time of 0.02 s between the scans). The fatty acids content was identified by comparing their retention times with those of known standards (37 components FAME Mix, Supelco no. 47885-U, Darmstadt, Germany) and the resulting mass spectra to those in the database (NIST MS Search 2.0). The amount of each fatty acid was expressed as a percentage of total fatty acid content.

#### 2.4.2. Crude Glycerol Analysis by FTIR

Crude glycerol fraction was analyzed by FTIR (Shimadzu IR Prestige-21, Kyoto, Japan) equipped with an attenuated total reflectance (ATR) module against petroleum ether as the background. The spectra were recorded on a wavelength range of 600–4000 cm^−1^ at a resolution of 4 cm^−1^ and 64 scans for a spectrum.

#### 2.4.3. Biomass and Cell Viability

Biomass growth was established by measuring the cell dry weight (CDW); 10 mL of the culture broth was filtered through 0.20 µm filters, which were further washed twice with double distilled water and dried at 104 °C for 8 h.

Yeast cell viability was determined by diluting 1 mL of fermentation sample in 9 mL of sterile saline solution (0.8% NaCl). Then, 100 µL of different dilutions were inoculated on yeast malt extract agar plates and incubated for almost 2 days at 30 °C. The viability of *C. zeylanoides* cells (log10 cfu/mL) was established by plate counting [51]. For microscopic examination of yeast cells, a loop of inoculated fermentation media was put on a glass laboratory lamella, dyed with methylene blue, and examined at 400× magnification [42,52].

#### 2.4.4. Organic Acids and Substrate Consumption (Glycerol, Glucose) Determination

Citric and succinic acids were determined using HPLC (Agilent 1200, Santa Clara, CA, USA) with an Aclaim OA (5 µm, 4 × 150 mm, Dionex, Waltham, MA, USA) reversed-phase chromatographic column coupled with UV detector, solvent degasser, quaternary pumps, column thermostat, and manual injector (Agilent Technologies, Santa Clara, CA, USA). The chromatographic column was eluted for 10 min with 50 mM NaH_2_PO_4_, pH 2.8, with a flow rate of 0.5 mL/min, at 20 °C. The chromatograms were measured at 210 nm.

Glycerol consumption was determined by derivatizing the sample after the method proposed by Imbert et al. [53], as presented in Figure 1, and then analyzed by HPLC. The analytical system consisted of an HPLC Agilent 1200 unit containing a quaternary pump, a solvent degasser, an autosampler, a UV-Vis photodiode detector (DAD) coupled with single quadrupole mass detector (MS, Agilent 6110), equipped with electrospray ionization source (ESI) (Agilent Technologies, California, Santa Clara, CA, USA), and controlled by Agilent ChemStation software. The ESI detection in positive ionization mode was done using the following work conditions: capillary voltage 3100 V, 350 °C, nitrogen flow 7 L/min, m/z 100–500 full-scan. The interest compounds separation was performed with an Eclipse XDB C18 column (5 μm, 4.6 × 150 mm I.D.) (Agilent Technologies, California, Santa Clara, CA, USA) using the 20 mM NH_4_HCO_2_ mobile phase (A), pH 2.8, and (B) CH_3_CN/NH_4_HCO_2_ (90/10, *v*/*v*) at a flow rate of 0.3 mL/min at 25 °C. The separation started with 50% B and increased up to 100% B for 10 min, and these conditions were maintained for 1 min. After 30 s, the original conditions were maintained for 15 min.

## 3. Results and Discussion

### 3.1. The Fatty Acids Profile from Processed and Unprocessed Vegetable Oil

The profile of fatty acids content from vegetable oil before and after being processed within the kitchen was identified. The results are presented in Figure 2. The recycled kitchen oil that was subjected to transesterification contained a high amount of (9Z,12Z)-octadeca-9,12-dienoic acid (74.74%). As can be seen from Figure 2, before being thermally processed (e.g., frying), the profile of fatty acids in vegetable oil showed an elevated concentration of (9Z)-octadecenoic acid (92.10%). During the frying, the concentration of the major compound decreased [(9Z)-octadecenoic acid], while the content of (9Z,12Z)-octadeca-9,12-dienoic acid increased. This might have been associated with the elevated temperatures of frying ranging from 130 to 180 °C, which could have facilitated the conversion of (9Z)-octadecenoic acid to (9Z,12Z)-octadeca-9,12-dienoic acid [18].

### 3.2. Crude Glycerol Obtaining Process

Crude glycerol was obtained from recycled kitchen oil through methanol transesterification in the presence of NaOH as the catalyst. The content of glycerol in the crude glycerol fraction was 30%, as determined by HPLC. The crude glycerol phase appeared as a dark yellow viscous solution at room temperature (23–25 °C) and as a compact solidified mass at low temperatures (<18 °C) (Figure 3).

The presence of glycerol in the crude glycerol phase was shown by FTIR spectra (Figure 4), where the functional groups associated with glycerol, ―OH bonds, were indicated by the presence of the large peak at 3358 cm^−1^. Moreover, the sharp peaks shown at 2924 and 2852 cm^−1^ represented C―H stretching that could be linked with the presence of polyalcohols, especially glycerol. The presence of bands at 1460 to 1436 cm^−1^ indicated C―O―H stretching connected with polyalcohols. The C―O groups associated with the presence of glycerol were suggested by the sharp peak at 1037 cm^−1^. In the crude glycerol phase, different quantities of impurities such as soaps, salts, or fatty acids were present. For instance, the high-pitched peak situated at 1741 cm^−1^ indicated the presence of C=O bonds specific to carboxylic acids or esters of fatty acids. The band at 1560 cm^−1^ showed the presence of COO^―^ groups particular to soaps or salts, which, in this case, may have been attributed to the catalyst involved in the transesterification reaction [54,55].

### 3.3. C. zeylanoides Growth in Different Culture Media

Numerous strains of *Candida* spp. have been industrially exploited for their beneficial usages in the food industry, such as *C. zeylanoides*, *Candida etchellsii*, *Candida intermedia*, *Candida maltosa*, and *Candida versatilis* [56]. Some of these microorganisms are well known as competent producers of organic acids, especially lactic, citric, isocitric, succinic, tartaric, or malic acids [47,57,58,59,60]. These strains are able to metabolize various sources of carbon (e.g., glycerol, glucose, fructose, galactose, mannose, mannitol, raffinose, ribose, and sucrose) [41], but some of them cannot metabolize other nutrient sources such as arabinose, cellobiose, maltose, melezitose, melibiose, rhamnose, salicin, trehalose, or xylose [42].

In this study, *C. zeylanoides* ATCC 20367 grew efficiently and developed full body cells in cultivation media that contained crude glycerol derived from recycled cooking oil, with the growth curves—namely viability and biomass formation—being similar to those obtained for pure glycerol or glucose as sole carbon sources. As is known from the literature [44,61], the growth of a particular yeast strain such as *Candida* or *Yarrowia* spp. is dependent on carbon concentration, nitrogen sources, aeration grade, and temperature. Moreover, when fermentation processes are conducted in stirred tank bioreactors, the mixing control plays an important role in cell growth due to the oxygen transfer rate to the fermentation broth [62]. In this context, for the present experimental work, the mixing was kept constant at 400 rpm, while a peristaltic pump was continuously bubbling filtered air (158 mL/min) in the cultivation broth for the entire fermentation process.

Even though the glycerol content in the crude glycerol fraction was low (30% in this case), the yeast cells grew successfully and developed full adult bodies after 48 h of cultivation in inoculum medium. The yeast cells from inoculum were observed under microscope light (Figure 5).

At the moment of inoculation, the yeast cell viability in crude glycerol medium was 6.53 ± 0.24 (log10) cfu/mL. By 90 h of cultivation, the biomass had doubled, and the yeast viability had reached 12.86 ± 0.62 (log10) cfu/mL. The viability and the biomass formation were not affected by the presence of possible impurities in the crude glycerol fraction. Moreover, as can be observed in Figure 6A, the cell viability and the biomass formation were similar to those obtained for pure glycerol (Figure 6B) and for glucose (Figure 6C) as main nutrient sources. In the case of yeast cells grown on crude glycerol, their viability reached the maximum point at 93 h of cultivation, which corresponds to complete glycerol consumption from the fermentation media.

The yeast cells viability (Figure 6A) slightly decreased until the end of the process, but the biomass quantity was still increasing after 163h of cultivation. The biomass formation yield of *C. zeylanoides* ATCC 20367 on crude glycerol reached a maximum value of 0.98 g/g after 163h, while the yields obtained for pure glycerol and glucose were 0.21 g/g and 0.31 g/g, respectively (Table 2). The biomass formation yield of this particular strain was higher as compared with other yeast species cultivated on biodiesel derived glycerol, such as a wild type of *Saccharomyces cerevisiae* that has recorded a maximum yield of 0.56 g_Biomass_/g_Substrate_ [63]. A comparison between the *C. zeylanoides* ATCC 20367 growth on crude glycerol, pure glycerol, and glucose is illustrated in Figure 7.

Considering the results registered for the biomass formation when glucose was used as a nutrient source, our results were significantly higher compared with those reported by Kamzolova and Morgunov (2017) [64], who cultivated three different species of *C. zeylanoides* (VKM Y-6, VKM Y-14, VKM Y-2324) on glucose. They achieved a maximum biomass concentration of 3.71 g/L after six days of fermentation when the strain *C. zeylanoides* VKM Y-6 was used [64], while in our experiment, the biomass concentration exceeded 8 g/L when *C. zeylanoides* ATCC 20367 was grown on glucose for almost six days (163 h) (Figure 6C).

The elevated yields for both biomass and citric acid when crude glycerol was used (Table 2) could have been attributed to the presence of fatty acids in crude glycerol fraction (Figure 4), which stimulated the enzymatic activity of the lipophilic yeast cells, generating a higher production of metabolites or biomass [65,66]. As Morgunov et al. (2013) [65] implied, the utilization of waste glycerol for the cultivation of *Y. lipolytica* strain NG40/UV7 increased the citric acid formation yield with 40.63% compared with the results obtained for pure glycerol [65].

### 3.4. Succinic and Citric Acids Bio-Synthesis by C. zeylanoides ATCC 20367

Both citric and succinic acids are small organic acids synthesized during TCA cycles by multiple microorganisms [26,60,65,66] and which can be obtained from crude glycerol by using microbes such as yeast and fungal strains [44,67,68]. Until now, little was known about the assimilation mechanism of the crude glycerol fraction by the yeast cells and the biosynthesis of organic acids. As Morgunov and Kamzolova (2015) stated [67], the crude glycerol fraction that consists of both glycerol and different amounts of fatty acids can be metabolized either together or separately. For the synthesis of organic acids from TCA cycle (e.g., citric and succinic acids) starting from crude glycerol as the main carbon source, specific enzymes are stimulated [65,69] such as glycerol kinase, isocitrate lyase, citrate synthase, aconitate hydratase, NAD- and NADP-dependent isocitrate dehydrogenases glycerol kinase, isocitrate lyase, citrate synthase, aconitate hydratase, and NAD- and NADP-dependent isocitrate dehydrogenases [67]. When glycerol or glucose is used as a nutrient source for yeast strains such as *Candida* or *Yarrowia*, many other metabolites can be synthesized (e.g., fumaric acid, pyruvic acid, α-ketoglutaric acid, erythritol, mannitol, etc.) by stimulating/inhibiting specific enzymes or by limiting particular biogenic microelements [64,66,69,70]. Moreover, the organic acids bio-production, especially citric and succinic acids, is closely related to air saturation and nitrogen-limited conditions when pH values are maintained over 4.5 [44,71,72]. In the present study, the biosynthesis of organic acids (citric and succinic acids) differed considerably because of the carbon source used, as is presented in Table 3, Table 4 and Table 5. The highest values of organic acids concentrations were observed for glucose (Table 5), while the lowest quantities were observed for crude glycerol (Table 3). Our results related to citric acid concentration were close to those reported by Mirończuk et al. [70], who obtained 6.7 ± 3.2 g/L in flask-shake cultivation and 1.4 ± 0.42 g/L at the batch bioreactor level by using different strains of *Y. lipolytica* on pure glycerol-containing media. The differences between the organic acids production when glycerol or glucose is used might be due to the fact that yeast strains such as *Candida* or *Rhodosporidium* [39,42] are glucophilic strains. The metabolism of glycerol versus glucose involves a different enzymatic package, which leads to the formation of organic acids [39,71].

The low content of glycerol in the crude glycerol fraction impacted the organic acids production by *C. zeylanoides* (Table 3).

Succinic acid is one of the organic acids with a major role as a monomer for the production of a number of polymers [73]. It is mostly bio-synthesized at high concentrations by different microorganisms from glucidic substrates such as glucose, fructose, lactose, maltose, etc. [73]. Even though the main carbon source is glycerol or glucose, the succinic acid biosynthesis by yeast strains such as *Candida* and *Yarrowia* is limited by the oxygen present in the culture medium, because a specific enzyme such as succinate dehydratase catalyzes the oxidation of succinate to fumarate [72]. Cultivated on pure glycerol, *C. zeylanoides* biosynthesized up to 15.66 g/L of succinic acid after 163 h of cultivation (Table 4). In our case, for the succinic acid biosynthesis by *C. zeylanoides* ATCC 20367, glucose was the appropriate substrate for increased concentrations production (Table 5) in a time interval of 75 h. After this interval, the succinic acid production abruptly decreased until the end of the process, a fact that could be explained by its conversion to other intermediates of the TCA cycle, organic acids such as fumaric, malic, or α-ketoglutaric acids [73,74]. Compared with literature, our results after 75 h of cultivating *C. zeylanoides* ATCC 20367 on media containing glucose were comparable with those achieved by Yuzbashev et al. [75], who cultivated a genetically engineered strain of *Y. lipolytica* on a mixture of glucose and glycerol. They obtained a maximum concentration of 45 g/L succinic acid after 168 h using shake flask fermentation [75].

In order to emphasize to the industry sector the terms of costs and the economic feasibility, we strongly recommend, considering the present findings, the co-production of biodiesel and valuable compounds such as citric and succinic acids in the one-pot synthesis process. Therefore, it is expected that biodiesel industries may better direct their by-product (crude glycerol), thus avoiding a growing environmental problem and providing an extra source of income.

## 4. Conclusions

Recycled kitchen oils are a cheap source of reusable biomass for the manufacturing of renewable biofuels such as biodiesel. The crude glycerol fraction that remains after transesterification constitutes a valuable nutrient source for microorganisms such as *C. zeylanoides*. In this study, *C. zeylanoides* ATCC 20367 was cultivated for 163 h on a crude glycerol-containing medium with a low content of glycerol, and its viability was maintained at high values until the glycerol content was depleted after 93 h. *Candida*’s viability and biomass formation rate were similar to those achieved for batch fermentations with pure glycerol or glucose as a single carbon source. In cultivation media containing crude glycerol, due to the lower quantity of glycerol, *C. zeylanoides* ATCC 20367 produced lower amounts of citric and succinic acids as compared with pure glycerol and glucose. The fact that this particular strain grows efficiently on lipophilic substrates such as raw glycerol derived from biodiesel represents a promising perspective for future investigations of crude glycerol fractions in order to obtain valuable chemicals (e.g., organic acids) with high yields.

The results presented within this paper contribute to the scientific background by highlighting the importance of some investigated parameters, such as viability. The elevated viability of the *C. zeylanoides* cells when it was cultivated on crude glycerol indicated that this particular strain is flexible and adapts its metabolism to the growing environment, leading to a step forward for future studies of yeast metabolism and their potential to synthesize valuable compounds. Moreover, the viability results from this research contribute to the fundamental academic research considering the utilization of yeast strains in biotechnological processes.

## Figures and Tables

**Figure 1 microorganisms-07-00265-f001:**
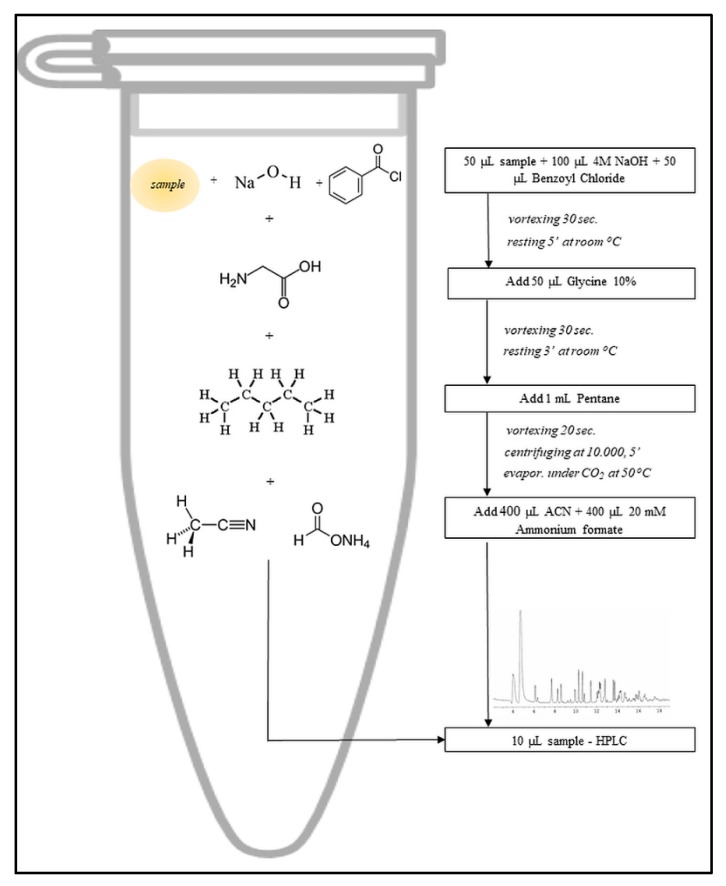
Samples derivatization flowchart adapted after Imbert et al. [53]. Glucose consumption was measured with an enzymatic test kit from Boehringer Mannheim-R-Biopharm. Biomass and metabolites production yields (Y) were calculated by using the formula [38]. Y (g/g) = Product concentration (g/L)/Initial substrate concentration (g/L).

**Figure 2 microorganisms-07-00265-f002:**
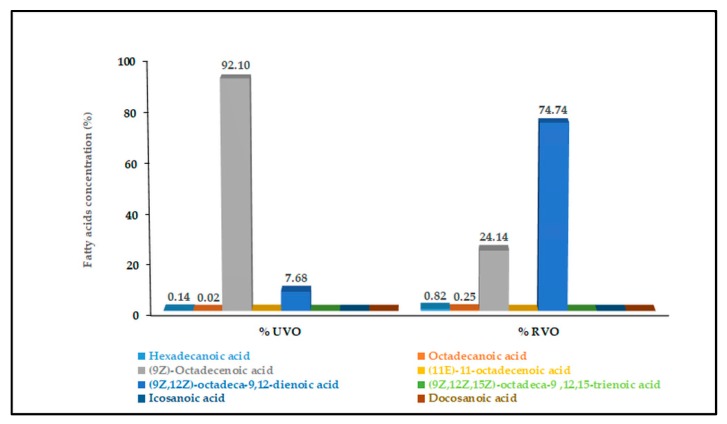
The fatty acids profile in vegetable oil before and after processing. UVO, unprocessed vegetable oil; RVO, recycled vegetable oil.

**Figure 3 microorganisms-07-00265-f003:**
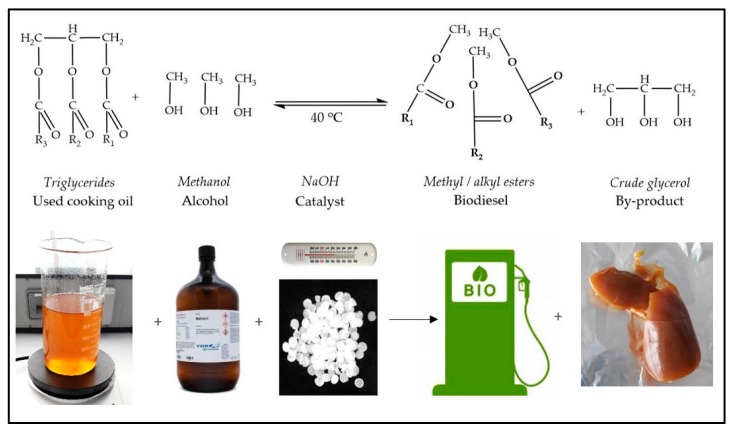
The crude glycerol obtaining process through transesterification.

**Figure 4 microorganisms-07-00265-f004:**
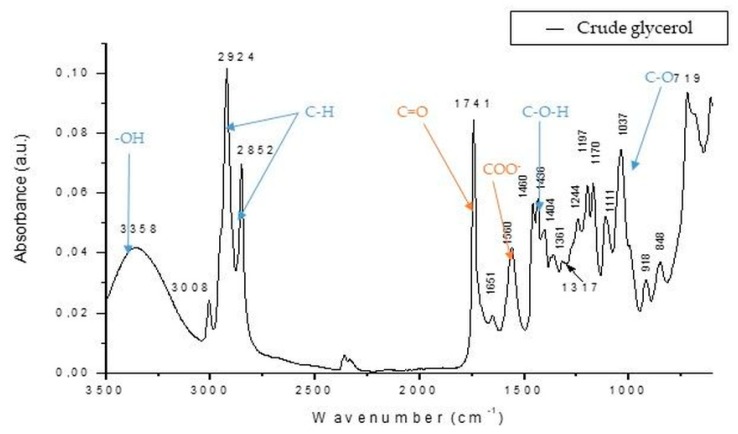
FTIR spectra of crude glycerol obtained from recycled kitchen oils through alkali transesterification. The blue bonds are linked with polyalcohols (glycerol) and the orange ones are associated with the impurities existing in the crude glycerol phase.

**Figure 5 microorganisms-07-00265-f005:**
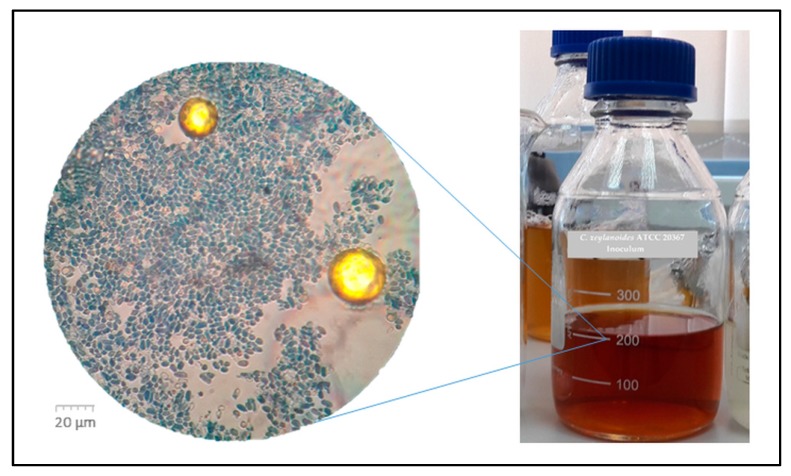
*C. zeylanoides* ATCC 20367 after 48 h in inoculum fermentation medium that contained crude glycerol under microscopic light. The blue stained dots represent viable cells, while the transparent shapes are the dead cells. The big yellow circles represent lipid droplets from the crude glycerol fraction.

**Figure 6 microorganisms-07-00265-f006:**
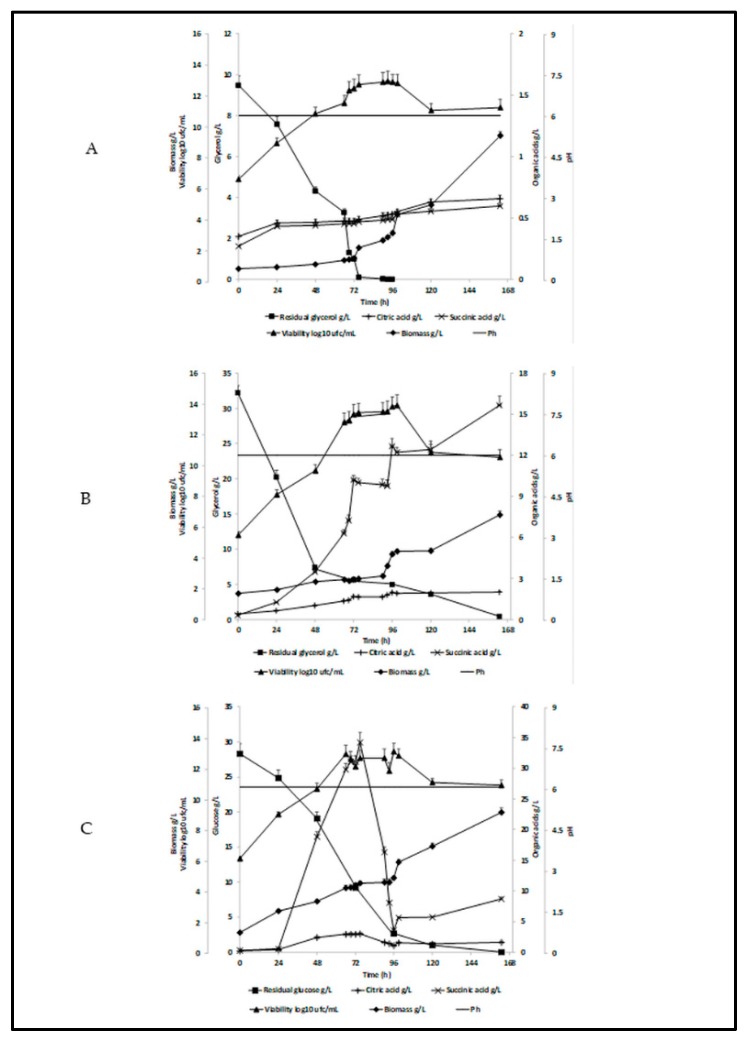
Yeast cell viability, biomass, pH, organic acids (citric, succinic), and substrate consumption on different carbon sources. (**A**) crude glycerol; (**B**) pure glycerol; (**C**) glucose.

**Figure 7 microorganisms-07-00265-f007:**
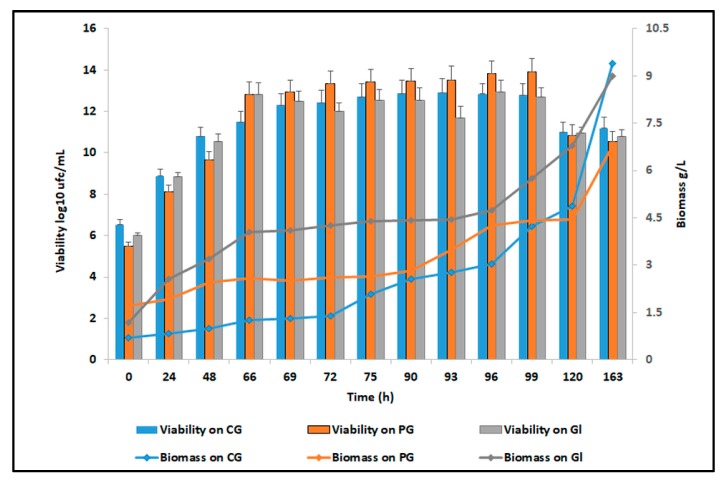
Comparison between the yeast cell growth (viability, biomass) on crude glycerol, pure glycerol, and glucose during 163 h of batch cultivation.

**Table 1 microorganisms-07-00265-t001:** Media components for *C. zeylanoides* ATCC 20367 growth on pure glycerol, crude glycerol, and glucose.

Nutrients	Batch CG	Batch PG	Batch Gl
n paraffin (mL/L)	50	50	-
Pure glycerol (g/L)	-	30	0.5
Crude glycerol (g/L)	30	-	
Glucose (g/L)	-	-	30
NH_4_Cl (g/L)	5	5	5
KH_2_PO_4_ (g/L)	0.5	0.5	0.5
MgSO_4_ (g/L)	0.5	0.5	0.5
CaCO_3_ (g/L) *	80	80	10
MnSO_4_ × 4H_2_O (mg/L)	2	2	2
ZnSO_4_ × 7H_2_O (mg/L)	2	2	2
FeSO_4_ × 7H_2_O (mg/L)	10	10	10
CuSO_4_ × 5H_2_O (µg/L)	50	50	50
Thiamine-HCl (µg/L) **	100	100	100

CG, crude glycerol; PG, pure glycerol; Gl, glucose. * was sterilized separately and added only in the inoculum media. ** was added to the fermentation broth after sterilization at 121 °C through sterile filtration (0.45 µm).

**Table 2 microorganisms-07-00265-t002:** The maximum yields of biomass and metabolites formation of *C. zeylanoides* ATCC 20367 grown on crude glycerol, pure glycerol, and glucose.

Yield g/g	Batch CG	Batch PG	Batch Gl
Y_Biomass_/_Substrate_	0.98	0.21	0.31
Y_Citric acid_/_Substrate_	0.06	0.06	0.05
Y_Succinic acid/Substrate_	0.06	0.48	1.2

CG—Crude Glycerol; PG—Pure Glycerol; Gl—Glucose.

**Table 3 microorganisms-07-00265-t003:** Results obtained for *C. zeylanoides* ATCC 20367 grown on crude glycerol at pH 6. The shown data represent the mean values of three biological replicates, and the standard deviation (±) is under 5%.

Time (h)	Viability Log10 ufc/mL	Biomass g/L	Citric Acid g/L	Succinic Acid g/L	Residual Glycerol g/L
0	6.53 ± 0.24	0.7 ± 0.02	0.35 ± 0.01	0.28 ± 0.00	9.48 ± 0.45
24	8.88 ± 0.34	0.82 ± 0.04	0.46 ± 0.02	0.43 ± 0.01	7.57 ± 0.37
48	10.80 ± 0.42	0.99 ± 0.05	0.47 ± 0.02	0.44 ± 0.02	4.31 ± 0.21
66	11.48 ± 0.51	1.25 ± 0.03	0.47 ± 0.02	0.45 ± 0.02	3.28 ± 0.16
69	12.30 ± 0.56	1.31 ± 0.04	0.47 ± 0.00	0.45 ± 0.02	1.30 ± 0.05
72	12.41 ± 0.62	1.40 ± 0.02	0.47 ± 0.01	0.46 ± 0.01	1.00 ± 0.05
75	12.70 ± 0.62	2.07 ± 0.01	0.49 ± 0.02	0.47 ± 0.01	0.10 ± 0.00
90	12.86 ± 0.62	2.56 ± 0.11	0.52 ± 0.02	0.48 ± 0.02	0.02 ± 0.00
93	12.90 ± 0.66	2.76 ± 0.10	0.52 ± 0.01	0.48 ± 0.02	0.00 ± 0.00
96	12.84 ± 0.51	3.03 ± 0.15	0.53 ± 0.00	0.49 ± 0.02	-
99	12.77 ± 0.57	4.24 ± 0.13	0.55 ± 0.02	0.53 ± 0.02	-
120	11.00 ± 0.46	4.86 ± 0.15	0.63 ± 0.02	0.55 ± 0.02	-
163	11.18 ± 0.55	9.38 ± 0.26	0.66 ± 0.02	0.60 ± 0.02	-

**Table 4 microorganisms-07-00265-t004:** Results obtained for *C. zeylanoides* ATCC 20367 grown on pure glycerol at pH 6. The shown data represent the mean values of three biological replicates, and the standard deviation (±) is under 5%.

Time (h)	Viability Log10 ufc/mL	Biomass g/L	Citric Acid g/L	Succinic Acid g/L	Residual Glycerol g/L
0	5.48 ± 0.21	1.70 ± 0.02	0.41 ± 0.02	0.33 ± 0.01	32.22 ± 1.01
24	8.11 ± 0.31	1.93 ± 0.05	0.65 ± 0.02	1.27 ± 0.02	20.22 ± 1.00
48	9.67 ± 0.40	2.45 ± 0.10	1.02 ± 0.03	3.48 ± 0.10	7.42 ± 0.35
66	12.83 ± 0.58	2.58 ± 0.11	1.34 ± 0.05	6.31 ± 0.23	-
69	12.92 ± 0.58	2.5 ± 0.12	1.40 ± 0.06	7.23 ± 0.38	-
72	13.32 ± 0.62	2.62 ± 0.12	1.66 ± 0.03	10.20 ± 0.33	5.60 ± 0.27
75	13.42 ± 0.62	2.65 ± 0.12	1.66 ± 0.03	9.99 ± 0.32	-
90	13.48 ± 0.61	2.83 ± 0.13	1.64 ± 0.05	9.83 ± 0.44	-
93	13.52 ± 0.66	3.48 ± 0.13	1.79 ± 0.06	9.75 ± 0.45	-
96	13.82 ± 0.62	4.25 ± 0.14	1.96 ± 0.07	12.66 ± 0.56	5.01 ± 0.22
99	13.91 ± 0.66	4.43 ± 0.11	1.93 ± 0.05	12.22 ± 0.35	-
120	10.84 ± 0.52	4.46 ± 0.15	1.95 ± 0.06	12.43 ± 0.61	3.61 ± 0.17
163	10.54 ± 0.48	6.80 ± 0.24	2.00 ± 0.07	15.66 ± 0.66	0.45 ± 0.01

**Table 5 microorganisms-07-00265-t005:** Results obtained for *C. zeylanoides* ATCC 20367 grown on glucose at pH 6. The shown data represent the mean values of three biological replicates, and the standard deviation (±) is under 5%.

Time (h)	Viability Log10 ufc/mL	Biomass g/L	Citric Acid g/L	Succinic Acid g/L	Residual Glucose g/L
0	6.00 ± 0.11	1.17 ± 0.02	0.23 ± 0.01	0.28 ± 0.00	28.35 ± 1.41
24	8.86 ± 0.17	2.57 ± 0.11	0.45 ± 0.01	0.59 ± 0.02	24.84 ± 1.18
48	10.53 ± 0.37	3.19 ± 0.12	2.41 ± 0.09	18.81 ± 0.80	19.06 ± 0.92
66	12.80 ± 0.58	4.06 ± 0.20	2.92 ± 0.10	29.76 ± 1.00	-
69	12.49 ± 0.49	4.09 ± 0.20	2.94 ± 0.12	31.02 ± 1.02	-
72	12.00 ± 0.42	4.25 ± 0.12	2.94 ± 0.14	31.03 ± 1.00	9.22 ± 0.32
75	12.53 ± 0.53	4.39 ± 0.11	3.01 ± 0.15	34.24 ± 1.52	-
90	12.53 ± 0.61	4.43 ± 0.22	1.59 ± 0.06	16.36 ± 0.80	-
93	11.70 ± 0.55	4.44 ± 0.20	1.42 ± 0.02	8.09 ± 0.20	-
96	12.95 ± 0.56	4.74 ± 0.11	1.07 ± 0.04	3.62 ± 0.10	2.66 ± 0.10
99	12.69 ± 0.44	5.76 ± 0.19	1.57 ± 0.04	5.67 ± 0.11	-
120	10.95 ± 0.27	6.80 ± 0.20	1.37 ± 0.05	5.71 ± 0.15	1.01 ± 0.02
163	10.77 ± 0.33	9.00 ± 0.29	1.63 ± 0.07	8.71 ± 0.28	0.04 ± 0.00

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
