# Peer review of "Biodiesel-Derived Glycerol Obtained from Renewable Biomass—A Suitable Substrate for the Growth of Candida zeylanoides Yeast Strain ATCC 20367"

_microorganisms, 2019, doi:10.3390/microorganisms7080265_

Round 1
Reviewer 1 Report
Referees Comments
on the manuscript entitled "Biodiesel-derived glycerol bbtained from renewable biomass – a suitable substrate for the srowth of Candida zeylanoides yeast strain ATCC 20367” for “Microorganisms”
Currently, the biodiesel-derived glycerol is widely studied as a carbon source for the development of processes of microbial synthesis of organic acids, lipids, proteins, etc. The authors obtained the crude glycerol from trans-esterified used kitchen oil and carried out the comparative analysis of crude glycerol, purified glycerol and glucose as a carbon source for the growth of Candida zeylanoides and organic acid biosynthesis. The great advantage of the article is the high methodological level of work. The results are presented with a logical sequence, beautifully illustrated. Of course, this work is of a high interest and in the scope of “Microorganisms” and could be considered for publication in this journal. But on my opinion, the minor revision of the manuscript is required.
Page 2, lines 36-42 - Please to reformulate the purpose of the work in a traditional manner. Page 3, Table 1 – Please to include the name of the first column of the Table 1, for example as “Nutrients” Page 5, lines 5-6 – Please to correct the product yields on the traditional symbols such as YX/S, YCA, YSA Page 6, Figure 2 - Please the correct the fatty acids in the IUPAC nomenclature. Section “ zeylanoides” growth in different culture media” - It is necessary to analyze your own data with literature data. So, Kamzolova and Morgunov (2017) showed the excellent growth (3.2 g/l) and citric acid biosynthesis (3.8 g/l) in Candida zeylanoides yeast cultivated in the medium with glucose. This article should be included in the discussion.
Kamzolova SV, Morgunov IG (2017) Metabolic peculiarities of the citric acid overproduction from glucose in yeasts Yarrowia lipolytica. Bioresour Technol 243:433-440.
Figure 6 must be presented in block on one page. The quality of the picture needs to be improved. Page 9, Table 2 - Please to correct the product yields on the traditional symbols such as YX/S, YCA, YSA Page 9, lines 3-9 – The authors obtained the high yields on biomass and citric acid using biodiesel-derived glycerol.
It should be noted that earlier, Morgunov et al. (2013) reported that the application of biodiesel-derived glycerol for Y. lipolyica yeast cultivation increased the citric acid production yield by 40.63 % as compared with that obtained on pure glycerol. This article should be included in the discussion of presented results.
Morgunov IG, Kamzolova SV, Lunina JN (2013) The citric acid production from raw glycerol by Yarrowia lipolytica yeast and its regulation. Appl Microbiol Biotechnol. 97(16):7387-97
Section “Succinic and citric acids bio-synthesis by C. zeylanoides ATCC 20367”
The results on the synthesis of citric and succinic acids must be explained. What is the possible stimulating mechanism of acid formation in yeast Candida zeylanoides? Why citric and succinic acids are produced, but not other metabolites of TCA cycle? Do you measure the nitrogen and other elements which can be limited the cell growth and stimulate the over-synthesis of citric acid? What is the possible pathway for tsuccinic acid synthesis on a medium glycerol and glucose? The metabolism of citric acid production on glycerin-containing waste was reported in the manuscript of Morgunov and Kamzolova (2015)
Morgunov IG, Kamzolova SV (2015) Physiologo-biochemical characteristics of citrate-producing yeast Yarrowia lipolytica grown on glycerol-containing waste of biodiesel industry Appl Microbiol Biotechnol 99(15):6443-6450.
Author Response
Dear Reviewer #1, we kindly thank you for revising our manuscript. We’ve considered your suggestions and the manuscript was amended accordingly. Please find below a detailed list of the implemented changes:
Comment: Page 2, lines 36-42 - Please to reformulate the purpose of the work in a traditional manner. Page 3, Table 1 – Please to include the name of the first column of the Table 1, for example as “Nutrients” Page 5, lines 5-6 – Please to correct the product yields on the traditional symbols such as YX/S, YCA, YSA Page 6, Figure 2 - Please the correct the fatty acids in the IUPAC nomenclature.
Answer: At the introduction section (page 2, lines 36-42) the purpose of the work is reformulated in the traditional way, as follows: ‘In the context of pollution reduction, crude glycerol derived from biodiesel production using recycled kitchen oils represents a suitable medium for cultivating microorganisms like Candida spp. The main purpose of this work was to evaluate the adaptation mechanisms of C. zeylanoides ATCC 20367 cells to a cultivation medium that contains only crude glycerol as energy source and their potential to bio-synthesize citric and succinic acids. Crude glycerol obtained from recycled kitchen oil by alkali transesterification was used as a single energy substrate, and analytically grade glycerol and glucose were used as comparators for cells’ viability and metabolites production.’
Thank you very much for your observation related to Table 1 from page 3. The first column from Table 1 was named ‘Nutrients’ as you suggested.
At lines 5-6 from page 5, the product yields were corrected on the traditional symbols such as YX/S, YCA, YSA.
The Figure 2 from page 6 was amended accordingly; the fatty acids were named in IUPAC nomenclature. The chart type from Figure 2 was changed in 3D clustered columns for better data observation.
Section “zeylanoides” growth in different culture media” - It is necessary to analyze your own data with literature data. So, Kamzolova and Morgunov (2017) showed the excellent growth (3.2 g/l) and citric acid biosynthesis (3.8 g/l) in Candida zeylanoides yeast cultivated in the medium with glucose. This article should be included in the discussion.Kamzolova SV, Morgunov IG (2017) Metabolic peculiarities of the citric acid overproduction from glucose in yeasts Yarrowia lipolytica. Bioresour Technol 243:433-440.
At section ‘C. zeylanoides growth in different culture media’ more information about the biomass formation on glucose substrate was added. The suggested reference (Kamzolova and Morgunov, 2017) was cited within the manuscript: ‘Considering the results registered for the biomass formation when glucose was used as a nutrient source, our results are significantly higher compared with those reported by Kamzolova and Morgunov 2017 [65] who cultivated three different species of C. zeylanoides (VKM Y-6, VKM Y-14, VKM Y-2324) on glucose. They achieved a maximum biomass concentration of 3.71 g/L after 6 days of fermentation when the strain C. zeylanoides VKM Y-6 was used [65], while in our experiment the biomass concentration exceeded 8 g/L when C. zeylanoides ATCC 20367 was grown on glucose for almost 6 days (163h) (Figure 6C).’
Figure 6 must be presented in a block on one page. The quality of the picture needs to be improved. Page 9, Table 2 - Please to correct the product yields on the traditional symbols such as YX/S, YCA, YSA.
As you suggested, Figure 6 was presented in a block on one page, and the quality of the picture was improved (TIFF format).
In Table 2 from page 9, the yields symbols were replaced with the traditional ones such as YX/S, YCA, YSA.
Page 9, lines 3-9 – The authors obtained the high yields on biomass and citric acid using biodiesel-derived glycerol. It should be noted that earlier, Morgunov et al. (2013) reported that the application of biodiesel-derived glycerol for Y. lipolyica yeast cultivation increased the citric acid production yield by 40.63 % as compared with that obtained on pure glycerol. This article should be included in the discussion of presented results.Morgunov IG, Kamzolova SV, Lunina JN (2013) The citric acid production from raw glycerol by Yarrowia lipolytica yeast and its regulation. Appl Microbiol Biotechnol. 97(16):7387-97
Thank you very much for your suggestion. The reference ‘Morgunov et al. (2013) was included in the discussion: ‘The elevated yields for both biomass and citric acid when crude glycerol is used (Table 2), can be attributed to the presence of fatty acids in crude glycerol fraction (Figure 4) which stimulates the enzymatic activity of the lipophilic yeast cells generating a higher production of metabolites or biomass [66, 67]. As Morgunov et al. 2013 [66] imply, the utilization of waste glycerol for the cultivation of Y. lipolytica strain NG40/UV7 increased the citric acid formation yield with 40.63% as compared with the results obtained for pure glycerol [66].’
Section “Succinic and citric acids bio-synthesis by C. zeylanoides ATCC 20367”.
The results on the synthesis of citric and succinic acids must be explained. What is the possible stimulating mechanism of acid formation in yeast Candida zeylanoides? Why citric and succinic acids are produced, but not other metabolites of TCA cycle? Do you measure the nitrogen and other elements which can be limited the cell growth and stimulate the over-synthesis of citric acid? What is the possible pathway for succinic acid synthesis on a medium glycerol and glucose? The metabolism of citric acid production on glycerin-containing waste was reported in the manuscript of Morgunov and Kamzolova (2015).
Morgunov IG, Kamzolova SV (2015) Physiologo-biochemical characteristics of citrate-producing yeast Yarrowia lipolytica grown on glycerol-containing waste of biodiesel industry Appl Microbiol Biotechnol 99(15):6443-6450.
Thank you very much for your observations. Some aspects from the ‘Succinic and citric acids bio-synthesis by C. zeylanoides ATCC 20367’ section were clarified by inserting the following paragraphs:
‘Up to now, few is known about the assimilation mechanism of the crude glycerol fraction by the yeast cells, and the bio-synthesis of organic acids. As Morgunov and Kamzolova 2015 stated [68], the crude glycerol fraction that consists of both glycerol and different amounts of fatty acids can be metabolized either together or separately. For the synthesis of organic acids from TCA cycle (e.g. citric and succinic acids) starting from crude glycerol as the main carbon source, specific enzymes are stimulated [66, 70] such as glycerol kinase, isocitrate lyase, citrate synthase, aconitate hydratase, NAD- and NADP-dependent isocitrate dehydrogenases glycerol kinase, isocitrate lyase, citrate synthase, aconitate hydratase, NAD- and NADP-dependent isocitrate dehydrogenases [68]. When glycerol or glucose is used as a nutrient source for yeasts strains like Candida or Yarrowia, many other metabolites can be synthesized (e.g. fumaric acid, pyruvic acid, α-ketoglutaric acid, erythritol, mannitol, etc.) by stimulating/inhibiting specific enzymes, or by limiting particular biogenic microelements [65, 67, 70, 71]. Moreover, the organic acids bio-production, especially citric and succinic acids, is closely related to air saturation and nitrogen-limited conditions when pH values are maintained over 4.5 [45, 72, 73].’
‘Even though the main carbon source is glycerol or glucose, the succinic acid bio-synthesis by yeast strains like Candida and Yarrowia is limited by the oxygen present in the culture medium, because a specific enzyme such as succinate dehydratase catalyzes the oxidation of succinate to fumarate [73].’
The production of other organic acids such as malic, pyruvic, fumaric, or ketoglutaric acids was not in our interest, so their quantification was not performed. In the present experiments, no elements limitation was applied for the overproduction of citric and succinic acids because our study was more a comparative one related to the growth of C. zeylanoides ATCC 20367 on different carbon sources.
The suggested reference (Morgunov and Kamzolova, 2015) was cited within the manuscript.
We would like to thank for the valuable and helpful comments, and for the positive feedback related to our study.
Reviewer 2 Report
The submitted contribution deals with the use of used kitchen oil for the production of biodiesel. The obtained glycerol (as transesterification byproduct) was integrated into a bioreactor cultivation process as a nutrient source for the growth of Candida zeylanoides ATCC 20367.
In general, the paper is solid and conclusion consistent with experimental results. Only some minor revisions are suggested :
Please increase the resolution of Figure 1 and Figure 6. Together with Table 3, 4 and 5 is would be nice to add a new figure in order to compare results obtained for C. zeylanoides ATCC 20367 grown on crude glycerol, pure glycerol and glucose
Finally, authors are kindly requested to check the manuscript for typos and mistakes in the text. Otherwise, a nice work recommended for publication in Microorganisms Journal.
Author Response
Dear Reviewer #2, we kindly thank you for revising our manuscript and for your positive feedback. We have considered your suggestions related to the quality of the images, so the resolution of Figure 1 and Figure 6 was increased. Moreover, a new figure was inserted (Figure 7) illustrating a comparison between the cells’ growth (biomass and viability) on crude glycerol, pure glycerol, and glucose. The manuscript was checked and mistakes were corrected.
Reviewer 3 Report
The authors have submitted a manuscript in which they analyze the use of glycerol derived from the transesterification of used kitchen oil as a nutrient source for the growth of Candida zeylanoides ATCC 20367. Cells’ viability, biomass production and biosynthesis of organic acids using this by-products were compared to those obtained with batch cultivations on pure glycerol or glucose as main nutrient substrates.
The topic of this paper is suitable for the scope of the Journal. The novelty is due to the use of crude glycerol obtained from recycled kitchen oil as a single energy substrate. In this regard, in the Introduction it is better to point out the originality of the work (Page 2 -lines 36-42).
In paragraph 2.1it is unclear the origin of the oil: which is the oil used (sunflower, olive, ...)? A mixture of oils? Where was it collected? Which is the temperature used during frying?
Paragraph 3.1: Revise English grammar, discuss more in detail the results
Figure 2: Check the use of significant digits, consider the use of a table instead of the histograms
Figure 4: Use the “.” Instead of “,” for Absorbance
Improve the quality of Figure 6: it is very difficult to read
Page 9 Lines 7-9: Discuss the results. Why biomass formation is higher? Put the reference not in subscript
Page 12 Lines 13-17: Discuss the results. Why your results are different? Are there other studies in literature?
Lines 18-22: I think that other studies are needed in order to state this. Revise English.
Author Response
Dear Reviewer #3, we kindly thank you for revising our manuscript and for your feedback. We have considered your suggestions and some changes were implemented into the manuscript, as it is explained below:
Comment: The topic of this paper is suitable for the scope of the Journal. The novelty is due to the use of crude glycerol obtained from recycled kitchen oil as a single energy substrate. In this regard, in the Introduction it is better to point out the originality of the work (Page 2 -lines 36-42).
Answer: Thank you very much for your observation. At the ‘Introduction’ section, page 2, the originality of the work is highlighted as follows:
‘In the context of pollution reduction, crude glycerol derived from biodiesel production using recycled kitchen oils represents a suitable medium for cultivating microorganisms like Candida spp. In this work, the main purpose of this work was to evaluate the adaptation mechanisms of C. zeylanoides ATCC 20367 cells to a cultivation medium that contains only crude glycerol as energy source and their potential to bio-synthesize citric and succinic acids. Crude glycerol obtained from recycled kitchen oil by alkali transesterification was used as a single energy substrate, and analytically grade glycerol and glucose were used as comparators for cells’ viability and metabolites production.’
In paragraph 2.1it is unclear the origin of the oil: which is the oil used (sunflower, olive, ...)? A mixture of oils? Where was it collected? Which is the temperature used during frying?
At the ‘2.1. Materials’ section the type of oil and its origin were mentioned: ‘The fried vegetable sunflower oil collected from the household was mixed with methanol and NaOH.’
Vegetable oil when is processed within the kitchen reaches temperatures ranging from 130-1800C; this aspect is mentioned in the manuscript at ‘3.1. The fatty acids profile from processed and unprocessed vegetable oil’ section. For the present study, it was not in our interest to analyze the frying temperature of the vegetable oil, but the usage of fried oil in the transesterification reaction.
Paragraph 3.1: Revise English grammar, discuss more in detail the results. Figure 2: Check the use of significant digits, consider the use of a table instead of the histograms.
At the ‘3.1. The fatty acids profile from processed and unprocessed vegetable oil’ section the English grammar was improved, and the results were discussed in detail. Moreover, the quality of Figure 2 from was improved. In Figure 2 only the significant digits were mentioned.
Figure 4: Use the “.” Instead of “,” for Absorbance.
At the Figure 4 “.” was used instead of “,” as you suggested.
Improve the quality of Figure 6: it is very difficult to read.
The quality of Figure 6 was increased in order to not be difficult to read.
Page 9 Lines 7-9: Discuss the results. Why biomass formation is higher? Put the reference not in subscript. Page 12 Lines 13-17: Discuss the results. Why your results are different? Are there other studies in literature? Lines 18-22: I think that other studies are needed in order to state this. Revise English.
Considering your suggestions related to ‘3. Results and Discussion’ section, more information was inserted into the manuscript:
‘Considering the results registered for the biomass formation when glucose was used as a nutrient source, our results are significantly higher compared with those reported by Kamzolova and Morgunov 2017 [65] who cultivated three different species of C. zeylanoides (VKM Y-6, VKM Y-14, VKM Y-2324) on glucose. They achieved a maximum biomass concentration of 3.71 g/L after 6 days of fermentation when the strain C. zeylanoides VKM Y-6 was used [65], while in our experiment the biomass concentration exceeded 8 g/L when C. zeylanoides ATCC 20367 was grown on glucose for almost 6 days (163h) (Figure 6C). The elevated yields for both biomass and citric acid when crude glycerol is used (Table 2), can be attributed to the presence of fatty acids in crude glycerol fraction (Figure 4) which stimulates the enzymatic activity of the lipophilic yeast cells generating a higher production of metabolites or biomass [66, 67]. As Morgunov et al. 2013 [66] imply, the utilization of waste glycerol for the cultivation of Y. lipolytica strain NG40/UV7 increased the citric acid formation yield with 40.63% as compared with the results obtained for pure glycerol [66].’
‘Up to now, few is known about the assimilation mechanism of the crude glycerol fraction by the yeast cells, and the bio-synthesis of organic acids. As Morgunov and Kamzolova 2015 stated [68], the crude glycerol fraction that consists of both glycerol and different amounts of fatty acids can be metabolized either together or separately. For the synthesis of organic acids from TCA cycle (e.g. citric and succinic acids) starting from crude glycerol as the main carbon source, specific enzymes are stimulated [66, 70] such as glycerol kinase, isocitrate lyase, citrate synthase, aconitate hydratase, NAD- and NADP-dependent isocitrate dehydrogenases glycerol kinase, isocitrate lyase, citrate synthase, aconitate hydratase, NAD- and NADP-dependent isocitrate dehydrogenases [68]. When glycerol or glucose is used as a nutrient source for yeasts strains like Candida or Yarrowia, many other metabolites can be synthesized (e.g. fumaric acid, pyruvic acid, α-ketoglutaric acid, erythritol, mannitol, etc.) by stimulating/inhibiting specific enzymes, or by limiting particular biogenic microelements [65, 67, 70, 71]. Moreover, the organic acids bio-production, especially citric and succinic acids, is closely related to air saturation and nitrogen-limited conditions when pH values are maintained over 4.5 [45, 72, 73]. In the present study, the bio-synthesis of the organic acids (citric and succinic acids) differed considerably because of the carbon source used, as it is presented in Tables 3, 4, and 5.’
We would like to thank for the valuable and helpful comments, and for the positive feedback related to our study.